# Exogenous salicylic acid, abscisic acid, and shikimic acid enhance drought tolerance in tea by modulating antioxidant defense and osmotic regulation

Shelina Akter Sheuli, Mir Sultanul Arafin, Sadia Sultana, Md. Afser Rabbi(ID), Ajit Ghosh(ID)*

Department of Biochemistry and Molecular Biology, Shahjalal University of Science and Technology, Sylhet, Bangladesh

* aghosh-bmb@sust.edu

## Abstract

Tea (*Camellia sinensis*), a widely cultivated crop across more than 50 countries, is highly vulnerable to drought stress, which impairs growth and reduces yield. This study examines the potential of exogenously applied salicylic acid (SA), abscisic acid (ABA), and shikimic acid (ShA) in enhancing drought tolerance in tea plants. Plants were pre-treated with SA, ABA, and ShA for 4 days, followed by 10 days of drought stress during which no water was provided by either foliar spray or soil irrigation. Leaf samples collected on day 15 were analyzed for various physiological and biochemical markers, including antioxidant enzyme activities (POD and GST), reactive oxygen species ($H_2O_2$) levels, and contents of total chlorophyll, carotenoids, and proline. Results showed that all treatments mitigated drought-induced physiological damage, reduced oxidative stress, and elevated antioxidant enzyme activity compared to untreated control plants under drought conditions. Treated plants also exhibited higher proline and chlorophyll levels, suggesting improved osmotic regulation and photosynthetic efficiency. Among the treatments, SA demonstrated the most pronounced enhancement of drought tolerance. To clarify the functional roles and relationships of 76 genes found through a literature review associated with the SA and ShA pathways under drought stress were examined using Gene Ontology (GO) enrichment, subcellular localization, and KEGG pathway analysis. These findings indicate that exogenous application of SA, ABA, and ShA enhances drought resilience in tea by strengthening antioxidant defenses and maintaining cellular integrity, presenting a promising approach to sustaining tea production in drought-prone areas. Future studies should explore the potential of combining these inducers with genetic or microbial strategies to further enhance drought resilience in tea plants.

**Data availability statement:** All relevant data are within the manuscript and its Supporting information files.

**Funding:** The author(s) received no specific funding for this work.

**Competing interests:** The authors have declared that no competing interests exist.

**Abbreviations:** SA, Salicylic acid; ABA, Abscisic acid; ShA, Shikimic acid; POD, Peroxidase; GST, Glutathione-S-transferase; ROS, Reactive oxygen species; AsA, Ascorbic acid; GSH, Glutathione; SOD, Superoxide dismutase, CAT, Catalase; APX, Ascorbate peroxidase; GR, Glutathione reductase; GPX, Glutathione peroxidase; GSSG, Glutathione disulfide; NADPH, Nicotinamide adenine dinucleotide phosphate; NPR1, Nonexpressor of pathogenesis-related genes 1; P5CS, Δ1-pyrroline-5-carboxylate synthetase; LEA, Late embryogenesis abundant; PYR/PYL/RCAR, Pyrabactin resistance 1/Pyrabactin resistance-like/Receptor for activated protein C kinase; PP2C, Protein phosphatase 2C; SnRK2, Sucrose non-fermenting 1-related protein kinase 2; CRD, Completely Randomized Design; TCA, Trichloroacetic Acid; $H_2O_2$, Hydrogen Peroxide; PMSF, Phenylmethylsulfonyl Fluoride; PMS, Phosphate-Buffered Saline; K-P, Potassium Phosphate; CDNB, 1-Chloro-2,4-Dinitrobenzene; ZIP, Basic Leucine Zipper (a type of transcription factor); MAPK, Mitogen-Activated Protein Kinase; ABP9, ABA-responsive bZIP 9.

## Introduction

Tea (*Camellia sinensis*) is a perennial evergreen shrub native to East Asia, primarily grown for its leaves, which are processed into various types of tea, including green, black, and oolong. This plant thrives in tropical and subtropical climates, requiring specific conditions such as well-drained soil, adequate rainfall, and moderate temperatures. Tea cultivation plays a significant role in the economies of many countries, particularly in regions like China, India, Bangladesh, Sri Lanka, and Kenya, sustaining millions of livelihoods [1,2]. However, the increasing impact of climate change, particularly global warming and rising non-agricultural water demands, poses a severe threat to tea production. Drought conditions are expected to significantly reduce tea growth, yield, and quality. Studies have shown that drought can reduce tea production by 14–33% and lead to plant mortality rates of 6–19% [3]. In Bangladesh, drought has been associated with reduced tea yields, which is critical since the country exports around 18 million kilograms of tea annually, contributing about 0.81% to its GDP [4,5]. Specific estates in Bangladesh experienced yield declines between 7.72% and 12.52% over three consecutive years due to drought [5]. Similarly, in China, drought conditions have been estimated to cause an 11% to 35% reduction in tea production, leading to fluctuations in tea prices and economic losses [6,7].

Drought stress disrupts the physiological and biochemical homeostasis of tea plants, primarily by causing oxidative stress due to an imbalance between reactive oxygen species (ROS) production and scavenging [8–10]. Accumulated ROS damages cellular membranes, proteins, and nucleic acids, leading to programmed cell death (PCD) and reduced biomass accumulation. To counteract this, plants activate a complex antioxidant defense system comprising both enzymatic components—superoxide dismutase (SOD), catalase (CAT), ascorbate peroxidase (APX), glutathione reductase (GR)—and non-enzymatic antioxidants such as ascorbic acid (AsA), glutathione (GSH), and flavonoids [10–12]. These systems work in tandem to detoxify ROS and maintain redox homeostasis. For instance, SOD catalyzes the dismutation of superoxide into hydrogen peroxide ($H_2O_2$), which is subsequently broken down by CAT and APX. GSH acts as a key redox buffer, converting to its oxidized form (GSSG) upon scavenging ROS, with regeneration mediated by NADPH-dependent GR activity [13].

Drought stress activates a complex network of phytohormones and secondary metabolites that orchestrate a range of physiological and biochemical adaptations necessary for plant survival. Central to this response are hormonal signaling pathways that regulate stomatal closure to minimize water loss and promote root system modification to facilitate water uptake from deeper soil layers. Among the key regulators, salicylic acid (SA), shikimic acid (ShA), and abscisic acid (ABA) play pivotal roles in modulating antioxidant defenses, osmotic adjustment, and stress-responsive gene expression, thereby enhancing plant resilience under drought conditions. SA is a phenolic phytohormone widely known for its role in plant defense and stress regulation. Under drought conditions, SA induces partial stomatal closure, thereby reducing transpiration while preserving photosynthetic efficiency [14–16]. It enhances the antioxidant defense machinery by upregulating SOD, CAT, and APX activities and

modulates osmotic adjustment through increased proline and glycine betaine accumulation. Moreover, SA boosts mineral uptake and stabilizes chlorophyll content, which is often degraded during drought. Recent research highlights its interaction with nitric oxide (NO) signaling pathways, further enhancing stress tolerance by modulating mitochondrial function and reducing ROS generation [17,18].

ShA, a key intermediate in the shikimate pathway, contributes to the biosynthesis of aromatic amino acids (phenylalanine, tryptophan, and tyrosine) and secondary metabolites such as flavonoids and lignin. These compounds are essential for structural integrity and ROS scavenging [19,20]. ShA supports osmotic balance under drought by increasing proline content via the activation of Δ1-pyrroline-5-carboxylate synthetase (P5CS), a key enzyme in proline biosynthesis [21]. Emerging studies suggest that ShA also plays a role in "metabolic priming"—the pre-activation of defense-related metabolic pathways, potentially preparing plants to respond more effectively to subsequent stress events [22]. ABA is another critical plant hormone that plays a significant role in stress responses, particularly drought. It induces stomatal closure through $Ca^2$-dependent signaling cascades, reducing water loss. Exogenous ABA application promotes the accumulation of osmolytes such as proline and soluble sugars, which help plants maintain cellular turgor and stabilize stress-responsive genes, such as dehydration-responsive element-binding (DREB) proteins and ABA-responsive elements (ABREs), which regulate genes encoding late embryogenesis abundant (LEA) proteins, dehydrins, aquaporins, and osmolyte biosynthesis, which protect cellular structures and modulate water balance [23]. ABA promotes root elongation and lateral root proliferation, enhancing water uptake under dry conditions. Recent findings also suggest that ABA modulates lipid metabolism and polyamine synthesis, reinforcing membrane integrity and reducing oxidative stress [24].

Despite extensive research on the roles of SA and ABA in plant stress physiology, their exogenous application in tea plants remains underexplored. Again, the potential of ShA as a drought stress mitigator in tea cultivation is scarcely studied. These gaps limit the understanding of how these compounds might enhance drought resilience in tea, a crop sensitive to water deficit and of high economic importance. The present study aims to evaluate and compare the effectiveness of exogenously applied SA, ShA, and ABA in enhancing the antioxidant defense system and biomass retention of tea plants under drought conditions. By assessing their influence on ROS-scavenging enzymes and physiological stability, this research seeks to identify promising biochemical treatments that can sustain tea productivity in water-limited environments, ultimately contributing to sustainable agronomic practices for drought-prone tea-growing regions. By integrating field-level data with controlled pot experiment outcomes, the study seeks to validate the reproducibility and robustness of the biochemical and physiological responses. This approach enhances the ecological relevance and practical applicability of the proposed drought mitigation strategies for sustainable tea cultivation under real-world agronomic conditions.

## Materials and methods

### Pot experiment setup

The laboratory-based pot experiment was carried out under a rain-exclusion shelter at the Department of Biochemistry and Molecular Biology, Shahjalal University of Science and Technology (SUST), Sylhet, Bangladesh (average ambient temperature ≈ 25 °C). Four-month-old seedlings of the tea cultivar BTS1 (BT1 × TV1), collected from the Lakkatura Tea Estate (24.911270°N, 91.888347°E), were transplanted into 16 plastic pots of equal size (15 cm in width and 23 cm in height) containing well-draining soil. Each pot was equipped with drainage holes and watered daily with 15 mL of distilled water. After 3 days of acclimatization, baseline data (Day 0) were recorded. A completely randomized design (CRD) with two replicates per treatment was followed. Plants were grouped into four treatments: control (distilled water), SA (4 mM), ShA (4 mM), and ABA (100 μM). The concentration of SA and ABA was selected based on the previous experiment [25,26]. The concentration of 4 mM ShA was employed in this study, which exceeds the concentration of 160 ppm typically used in previous studies where seeds were soaked in the inducer solution [27]. Inducers were applied only once daily, in the early morning, by foliar spraying for four consecutive days (Days 0–3) at a single defined growth stage. Control plants received an equal volume of distilled water. Days 4 and 5 involved only water spraying without inducer application. From Day 6 onward,

drought stress was induced by withholding water for 10 days in the treatment groups. Control plants were maintained under regular watering (15 mL per day). Leaf samples were harvested on Day 15 for biochemical and enzymatic analyses.

## Field experiment

To validate the physiological and biochemical responses observed under controlled conditions, a parallel field-based experiment was conducted. For this experiment, four-month-old BTS1 plants were collected from Lakkatura Tea Estate (24.911270°N, 91.888347°E). The entire experiment was carried out in the nursery of Tarapur Tea Estate (24.9202° N, 91.8561° E). It was conducted in two independent raised soil beds to ensure isolation of treatments. One bed was designated for drought-treated plants (irrigation withheld), and the other for well-watered control plants, thereby preventing any water interference between treatments. Plants were transplanted into the beds with a spacing of 25 cm × 25 cm. Each treatment included three replicates, with a total of 24 plants used in the experiment. The experimental design mirrored that of the pot trial, with identical treatment groups ([SA], [ShA], [ABA], and control), concentrations (4 mM SA, 4 mM ShA, and 100 μM ABA), application schedules, and drought imposition timelines. Foliar applications were performed once daily during the early morning hours from Day 0 to Day 3. To stimulate drought stress while preserving exposure to natural environmental conditions, a rain-exclusion structure was installed, following established methodologies from previous field-based drought studies [28]. This setup enabled drought induction (from Day 6 to Day 15) by suspending irrigation while allowing plants to be exposed to natural light, temperature, and microclimatic variations such as fluctuating humidity and wind (Table 1). Leaf samples were collected on Day 15 for comparative analysis with the pot experiment to assess the reproducibility and ecological validity of the physiological and biochemical responses.

## Estimation of photosynthetic pigments

Chlorophyll "a" and "b" content was extracted and measured (in μg/ml FW) using a UV–Vis recording spectrometer at 663 and 646 nm following the method described previously [29]. Distilled water was mixed with reagent-grade acetone in a ratio of 2:8 for 80% acetone. Fresh leaf samples were collected and homogenized using a mortar and pestle with fine sand. 1.5 ml of 80% acetone was used as the extractant solvent. The homogenized sample was then transferred to Eppendorf tubes and centrifuged for 15 minutes, 4000 g at 4°C. After centrifugation, the supernatant was collected, which was then analyzed for Chlorophyll-a, Chlorophyll-b, and carotenoid content by detecting their absorbance with a UV-visible light spectrophotometer. All samples being analyzed were read at 646 nm, 663 nm, and 470 nm [29].

## Measurement of proline content

Proline contents were quantified by using the previously described protocol [30]. 200 mg of ground fresh leaf tissue was homogenized in 3 ml of 100 mM phosphate buffer ($NaH_2PO_4$; $NaHPO_4$), pH 7.8. The homogenate was centrifuged, and

**Table 1. Treatments in the pot experiment.**

| SI no | Symbol | Treatment |
|---|---|---|
| 1 | Control | Foliar spray of distilled water |
| 2 | Drought | Drought stress control (non-treated seedlings under drought stress) |
| 3 | ABA | Abscisic acid control (Foliar spray of ABA) |
| 4 | ABA+D | Plants sprayed with ABA under 10 days drought stress |
| 5 | SA | SA control (Foliar spray of SA) |
| 6 | SA+D | Foliar spray of SA under 10 days drought stress |
| 7 | ShA | ShA control (Shikimic acid foliar spray) |
| 8 | ShA+D | Plants sprayed with shikimic acid under 10 days drought stress |

the supernatant was stored as a total crude extract at −20°C. To estimate the proline content, 50 μl of crude mix was added to 1 ml of reaction mixture containing 250 μl of 3% sulphosalicylic acid, 250 μl of acetic acid, and 500 μl of 2.5% ninhydrin solution. The mixture was boiled in a boiling water bath for 15 min and cooled down on ice for 5 min. The absorbance value was read at 520 nm. The proline content was estimated using a standard curve prepared with varying ranges (0–1 mg/ml) of proline concentration.

## Measurement of hydrogen peroxide content

Levels of hydrogen peroxide from tea leaves were quantified by following the previously described protocol [31]. The leaf samples were weighed, homogenized with 1 ml of 0.1% trichloroacetic acid (TCA), and centrifuged. The leaf extract supernatant (200 μl) was mixed with 300 μl of 100 mM potassium phosphate buffer ($KH_2PO_4$, $K_2HPO_4$, pH 7.8) and 500 μl of reagent (1 M KI). The blank control consisted of 200 μl of 0.1% TCA, 300 μl of 100 mM potassium phosphate buffer (pH 7.8), and 500 μl of reagent (1 M KI). After 1 hour of reaction in the dark, the absorbance was measured at 390 nm. The amount of $H_2O_2$ was calculated using a standard curve prepared with known concentrations of $H_2O_2$.

## Extracting total protein for enzyme activity

To conduct enzyme activity from the total protein extract, proteins were isolated in their native form with appropriate care. All the steps should be carried out at 4°C. Fresh plant tissue (~200 mg) was ground and homogenized in an ice-cold extraction buffer containing 100 mM potassium phosphate buffer, pH 7.0, 50% glycerol, 16 mM $MgSO_4$, and 0.5 mM PMSF [32]. The homogenate is centrifuged at 12,000 g for 30 minutes at 4°C. The supernatant fraction is employed as a crude extract and quantified by the Bradford method [33].

## Peroxidase activity

Peroxidase (POD, EC: 1.11.1.7) activity was assessed according to the previous method [34]. The reaction mixture contained K-P buffer (25 mM, pH 7.0), guaiacol (0.05%), $H_2O_2$ (10 mM), and the enzyme solution (20 μL) (final volume 1000 μL). An increase in absorbance was recorded at 470 nm for 2 min, and the extinction coefficient of 26.6 mM$^{-1}$ cm$^{-1}$ was used for the calculation of activity.

## Glutathione S-transferase (GST) activity

GST activity was determined spectrophotometrically. The reaction mixture contained 100 mM Tris-HCl buffer (pH 6.5), 1.5 mM GSH, 1 mM 1-Chloro-2,4-dinitrobenzene (CDNB), and 20 μl protein solution in a final volume of 1 ml. The increase in absorbance was measured at 340 nm for 2 minutes. The activity was calculated using the extinction coefficient of 9.6 mM$^{-1}$ cm$^{-1}$ [34].

## Relative water content

The relative water content (RWC) of tea leaves was determined according to previous studies [35]. Fresh weight (FW), dry weight (DW), and fully turgid weight (FTW) of fully turgid weight were recorded. Relative water content was calculated by the following formula:

$$RWC\ (\%) = [(FW - DW)/(FTW - DW)] \times 100$$

## Subcellular localization, KEGG pathway, and gene enrichment analysis

A total of 76 genes (S1 Table) were selected through an extensive literature review to investigate their involvement in salicylic acid (SA) and shikimic acid (ShA) signaling pathways under drought stress conditions. The gene sequences

were retrieved from the NCBI database. To analyze these genes, data were generated using the Database for Annotation, Visualization, and Integrated Discovery (DAVID) (https://davidbioinformatics.nih.gov/) (S1 Table). A circular diagram was constructed using the web-based tool Chiplot (https://www.chiplot.online/). Additionally, a chord diagram was generated with OriginLab (https://www.originlab.com/). Furthermore, a dot plot denoting subcellular localization and the KEGG Pathway of the selected genes was created using STRING (https://string-db.org/).

## Statistical analysis

The data were subjected to a thorough statistical analysis to evaluate mean differences across various treatment groups. Initially, the variances of the samples were assessed using the F-test. This step was crucial for ensuring that the assumptions of the subsequent statistical tests were met. Following the F-test, an independent one-tailed Student's t-test was employed to compare the mean values of the control group against those of the different treatment groups. To further ascertain whether there were significant differences in the percentage changes among the treatments, the Least Significant Difference (LSD) method was applied. This method allows for the identification of specific treatment pairs that exhibit significant differences while maintaining an overall significance threshold of $P < 0.05$, as predetermined in the study design. This multi-step analytical approach ensures robust conclusions regarding the effects of various treatments on the measured outcomes.

## Ethics approval and consent to participate

All experimental research on plants, including the collection of plant materials, has been conducted by following the relevant institutional, national, and international guidelines and legislation. Permission to use the field area for the experiment was granted by the Manager of Tarapur Tea Estate, Rinku Chakrabarti.

## Result

### Exogenous treatments reduce drought-induced morphological damage

A marked difference in the physical condition of plants when subjected to drought stress, both with and without the application of specific chemical treatments (Fig 1). Before the imposition of drought stress, plants showed no visible signs of physical damage, which suggests a baseline of normal physiological condition across all groups (Fig 1, upper panel). However, the imposition of drought stress alone led to substantial physical deterioration. These changes were primarily observed in their leaf water content, color, and structural integrity, which are often indicators of overall plant health and drought resilience (Fig 1, lower panel). In contrast, plants that received treatments with SA, ABA, or ShA either alone or in combination with drought stress, exhibited significantly less physical damage (Fig 1, lower panel). This result highlights the protective role of these treatments against drought stress. The treated plants subjected to drought stress showed minimal and largely negligible changes in leaf color and structure when compared to their untreated, drought-stressed counterparts. The minimal physical damage observed in treated plants suggests that SA, ABA, and ShA may contribute to enhanced drought tolerance by mitigating some of the adverse effects of water deficit on plant physiology.

### Exogenous treatments safeguarded photosynthetic pigments

To investigate the protective role of exogenous SA, ABA, and ShA on photosynthetic performance under drought stress, total chlorophyll and carotenoid contents were quantified in tea leaves under both laboratory (pot) and field conditions. Drought-stressed, untreated plants exhibited a pronounced reduction in total chlorophyll content—20.00% in the pot (Fig 2A) and 17.01% in the field (Fig 2B) conditions from day 0, emphasizing the adverse effects of water deficit on pigment stability. In contrast, drought-stressed plants treated with SA, ABA, and ShA showed notable increases of 15.45%, 10.29%, and 13.00% in the pot (Fig 2A), and 19.30%, 9.99%, and 11.23% in the field (Fig 2B), respectively. Non-stressed

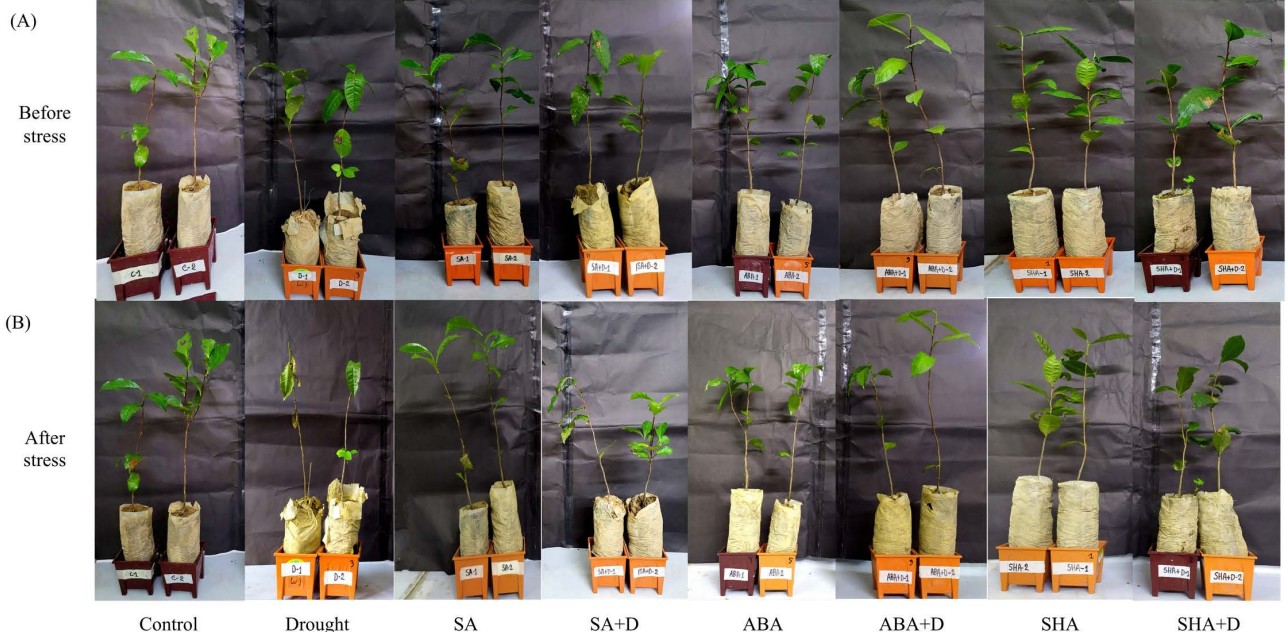

**Fig 1. Condition of plans before and after drought stress.** Before drought stress, all plants exhibited healthy leaves, vibrant color, and structural integrity, indicative of a baseline normal physiological state. No visible signs of stress or physical damage were observed, reflecting comparable initial conditions across both treated and untreated groups before the imposition of drought stress. Drought stress led to notable physical deterioration in untreated plants, as evidenced by reduced leaf water content, color loss, and compromised structural integrity. Plants treated with salicylic acid (SA), abscisic acid (ABA), and shikimic acid (ShA) displayed significantly less physical damage under drought stress conditions, suggesting enhanced drought resilience due to the protective effects of these treatments.

plants treated with the same compounds also exhibited enhanced chlorophyll accumulation, increasing by 38.41%, 18.32%, and 31.81% from day 0 in laboratory conditions (Fig 2A), with a similar pattern observed in the field (Fig 2B). No significant changes were observed in the untreated, well-irrigated control group, indicating pigment levels remained stable in the absence of both drought and treatment. A similar trend was observed for carotenoids for both laboratory (Fig 2C) and field experiments (Fig 2D). The untreated drought-stressed group experienced declines of 16.98% in laboratory conditions and 15.70% in field conditions from day 0. However, drought-stressed plants treated with SA, ABA, and ShA exhibited carotenoid increases of 13.35%, 8.75%, and 12.06% in the pot, and 16.31%, 8.12%, and 9.91% in the field, respectively. Likewise, non-stressed treated plants demonstrated substantial increases in carotenoid levels, 35.00%, 16.15%, and 24.71% in the pot experiment, with comparable results observed under field conditions. The untreated, non-stressed control plants showed no significant variation in carotenoid content.

## Exogenous application of plant hormones mitigates drought-induced damage through enhanced proline levels

To evaluate the impact of exogenous compounds on drought-induced oxidative stress mitigation, proline, a key Osmo protectant and stress biomarker, was quantified in tea leaves under both pot (Fig 3A) and field experiments (Fig 3B). All three treatments induced a significant increase in proline accumulation in drought-stressed plants compared to untreated stressed controls. Under laboratory (pot) conditions, SA, ABA, and ShA treatments resulted in increases of 63.46%, 55.23%, and 52.81%, respectively, relative to day 0 levels. Corresponding increases in the field experiment were 49.28%, 44.28%, and 41.73%, respectively. In contrast, untreated drought-stressed plants showed lower proline elevation: 40.37% in the pot and 34.48% in the field. These results underscore the enhanced efficacy of all three compounds in promoting

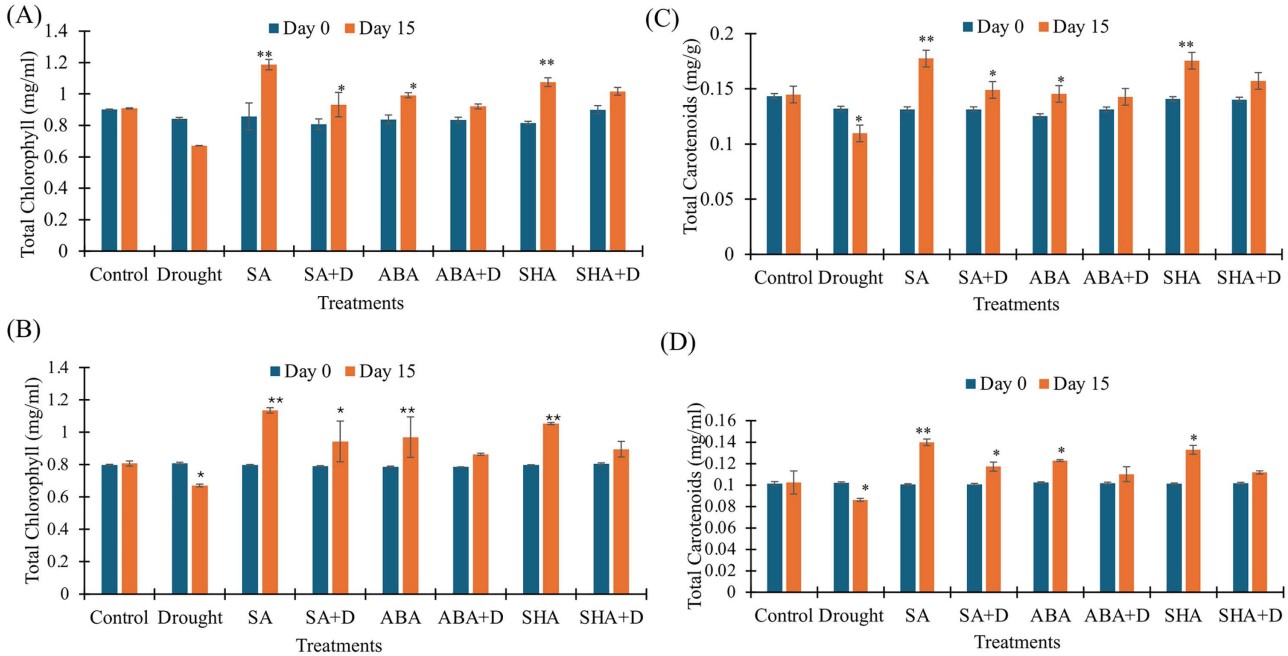

**Fig 2. Effect of Treatments on photosynthetic pigments in tea plants measured at Day 0 and Day 15.** A) Total Chlorophyll content (pot), B) Total Chlorophyll content (Field), C) Total Carotenoid content (pot), D) Total Carotenoid content (field). Treatments include salicylic acid (SA), salicylic acid with drought (SA+D), abscisic acid (ABA), abscisic acid with drought (ABA+D), shikimic acid (ShA), shikimic acid with drought (ShA+D), and drought alone. Concentration is expressed in mg/g. All values from the pot experiment represent the mean of the two replicates ± standard error of the mean (n = 3), values from the field experiment represent the mean of the three replicates ± standard error of the mean (n = 4). The significance level of the paired t-test is denoted by (*) and (**) with a p-value less than 0.05 and 0.01, respectively. Day 0 (blue); Day 15 (orange).

proline biosynthesis under stress conditions. Interestingly, even in the absence of drought, treated plants exhibited modest proline elevations—17.71% (SA), 14.38% (ABA), and 11.13% (ShA) in laboratory settings, with similar patterns in field trials. This suggests a priming effect, wherein treatments initiate stress-responsive pathways even under non-stressful conditions. No significant change was observed in proline content in the untreated, non-stressed control group across both environments, confirming the specificity of the observed responses to treatment and/or stress.

## Exogenous application of plant hormones reduces ROS level

To assess the efficacy of exogenous treatments in mitigating oxidative stress in tea plants under drought conditions, hydrogen peroxide ($H_2O_2$) levels were measured as an indicator of ROS accumulation. $H_2O_2$ content, which typically rises under drought-induced oxidative stress, was quantified in both pot (Fig 3C) and field experiments (Fig 3D). Drought-stressed plants treated with SA, ABA, and ShA showed significantly lower increases in $H_2O_2$ content from their respective day 0 levels compared to untreated drought-stressed controls. Under laboratory (pot) conditions, SA, ABA, and ShA treatments led to $H_2O_2$ increases of 21.36%, 23.20%, and 27.95%, respectively. In the field, the corresponding increases were 22.66%, 27.39%, and 28.36%. By contrast, the untreated drought-stressed plants exhibited markedly higher increases in $H_2O_2$—49.04% in the pot experiment and 45.79% in the field from day 0 levels, indicating greater oxidative damage. These results underscore the protective role of the exogenous treatments, which substantially limited $H_2O_2$ accumulation under drought stress, suggesting improved management of oxidative stress in treated plants. In non-stressed but treated plants, slight increases in $H_2O_2$ levels were also observed from day 0, 10.87% with SA, 8.37% with ABA, and 7.73% with ShA in laboratory conditions, and similar trends were noted in the field. These modest elevations are not associated

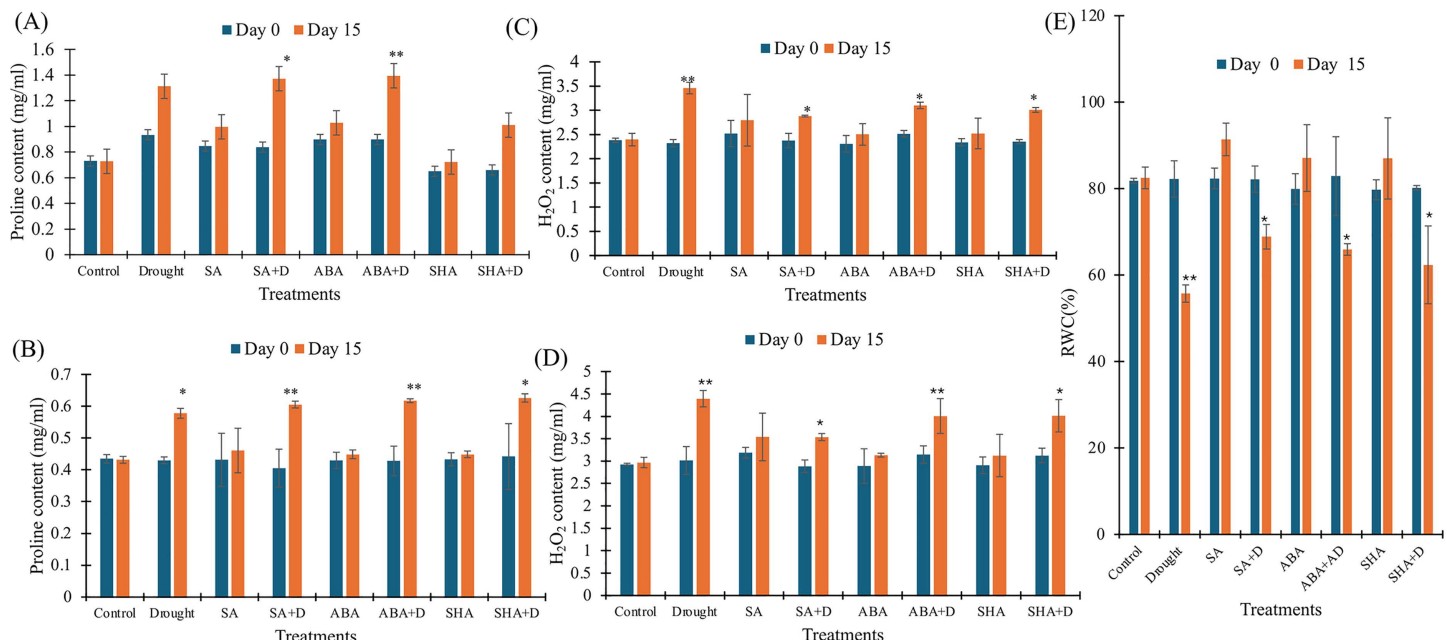

**Fig 3. Effect of treatments on non-enzymatic stress biomarkers in tea plants measured at Day 0 and Day 15.** A) Total Proline Concentration (pot), B) Total Proline Concentration (field), C) Total Hydrogen Peroxide Concentration (pot), D) Hydrogen Peroxide Concentration (field), E) Relative water content (field). Treatments include salicylic acid (SA), salicylic acid with drought (SA+D), abscisic acid (ABA), abscisic acid with drought (ABA+D), shikimic acid (ShA), shikimic acid with drought (ShA+D), and drought alone. Concentration is expressed in mg/ml. All values from the pot experiment represent the mean of the two replicates ± standard error of the mean (n = 3), values from the field experiment represent the mean of the three replicates ± standard error of the mean (n = 4). The significance level of the paired t-test is denoted by (*) and (**) with a p-value less than 0.05 and 0.01, respectively. Day 0 (blue); Day 15 (orange).

with oxidative damage but may reflect a priming or signaling role, as low-level ROS production can activate downstream stress-responsive and defense-related pathways. Untreated, non-stressed control plants showed no significant change in $H_2O_2$ levels from day 0, reinforcing that the observed effects were treatment-specific.

## Exogenous application of plant hormones maintains leaf water status during drought

Relative Water Content, a key physiological indicator of plant water balance, was assessed to determine how the treatments influence water retention under drought conditions in a field experiment (Fig 3E). Under drought stress, RWC declined sharply in untreated plants, showing a −32.48% deviation from day 0. In contrast, plants treated with SA, ABA, and ShA under drought conditions exhibited notably smaller reductions in RWC, with declines of −17.44%, −21.73%, and −26.83%, respectively. These values reflect a clear improvement in water retention compared to the untreated drought-stressed group, suggesting that the exogenous treatments mitigate water loss and contribute to drought resilience.

In non-stressed plants, treatment with SA, ABA, and ShA led to moderate increases in RWC from day 0 levels of 10.76%, 7.87%, and 7.96%, respectively, indicating that these compounds can enhance water status even in the absence of drought. This effect is likely linked to improved osmotic regulation or stomatal behavior. The untreated non-stressed control showed a minimal change of +0.86% in RWC, confirming that the observed effects are treatment dependent.

## Exogenous application of plant hormones enhances antioxidant defense mechanisms

Peroxidase (POD) activity remained largely unchanged in the control (well-watered, untreated) groups throughout the experimental period, with negligible variations recorded in both pot (Fig 4A) (1.30%) and field (Fig 4B) (−0.13%)

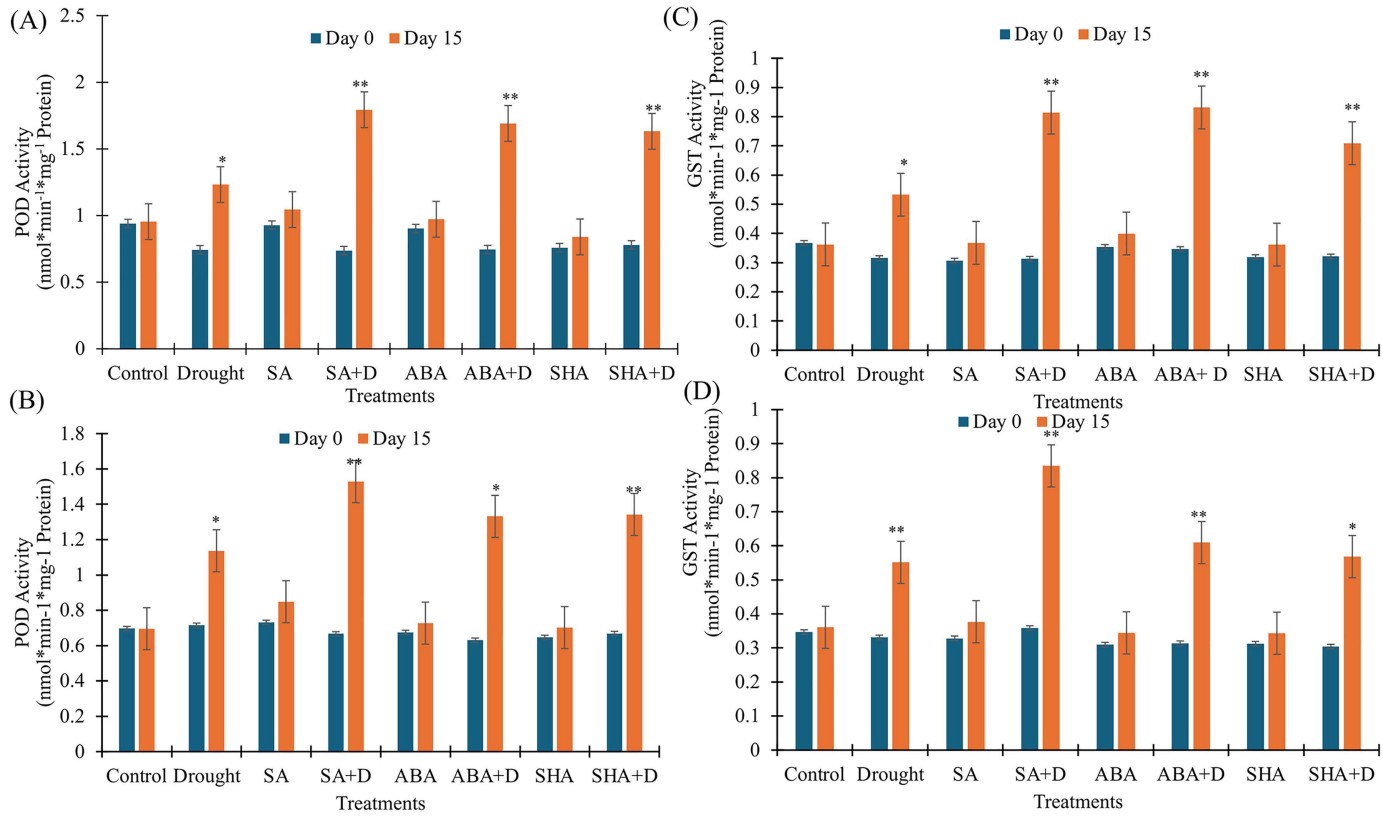

**Fig 4. Effect of treatments on the enzymatic stress markers activity in tea plants measured at Day 0 and Day 15.** A) Peroxidase Activity (pot), B) Peroxidase Activity (field), C) glutathione S-transferase Activity (pot), D) glutathione S-transferase Activity (field). Treatments include salicylic acid (SA), salicylic acid with drought (SAD), abscisic acid (ABA), abscisic acid with drought (ABA D), shikimic acid (ShA), shikimic acid with drought (ShA D), and drought alone. POD and GST activity is expressed in μmol/min/mg protein. All values from the pot experiment represent the mean of the two replicates ± standard error of the mean (n = 3), values from the field experiment represent the mean of the three replicates ± standard error of the mean (n = 4). The significance level of the paired t-test is denoted by (*) and (**) with a p-value less than 0.05 and 0.01, respectively. Day 0 (blue); Day 15 (orange).

conditions, indicating a stable baseline of enzymatic function under non-stressed conditions. Similarly, well-watered plants treated with SA, ABA, and ShA showed only minor increases in POD activity, suggesting limited induction of the antioxidant defense system under favorable environmental conditions. However, a markedly different trend was observed under drought stress. The application of elicitors significantly elevated POD activity, highlighting an enhanced antioxidant response. The SA-treated drought-stressed plants (SA + D) exhibited the highest increases in POD activity, rising by 143.46% in pot and 129.27% in field conditions. Corresponding increases were also noted for ABA-treated plants (127.28% pot, 111.04% field) and ShA-treated plants (109.34% pot, 100.73% field). In contrast, drought-stressed plants without elicitor treatment showed only modest increases of 65.78% (pot) and 58.87% (field).

To further evaluate the effect of exogenous elicitors on detoxification capacity, glutathione S-transferase (GST) activity was measured under non-stressed and drought conditions in both pot (Fig 4C) and field (Fig 4D) experiments. GST, a key antioxidant enzyme involved in ROS detoxification, remained relatively stable in the untreated, non-stressed control, with minor decreases of 1.43% (pot) and 4.04% (field). In contrast, well-watered plants treated with SA, ABA, and ShA showed moderate increases in GST activity, with laboratory-grown plants showing increases of 19.73%, 12.92%, and 13.38%, respectively, and field-grown plants showing increases of 14.80%, 11.11%, and 9.80% from Day 0. Under

drought stress, GST activity was significantly elevated in treated plants, reflecting a robust detoxification response. The greatest enhancement was again observed in SAD-treated plants, with GST activity rising by 159.82% (pot) and 132.92% (field). Treatments with ABA and ShA under drought also resulted in substantial increases—139.81% and 94.32% for ABA, and 120.78% and 86.94% for ShA in lab and field conditions, respectively. In contrast, drought-stressed plants without any treatment showed only modest increases of 68.74% (pot) and 66.63% (field).

### Salicylic acid outperforms abscisic acid and shikimic acid as an inducer

To evaluate elicitor efficacy under non-stressed conditions, percent changes in key biochemical parameters were assessed at day 15 post-treatment relative to untreated controls. All inducers improved chlorophyll and carotenoid content in both field (Fig 5A), and pot (Fig 5B) experiments, with SA showing the most pronounced effect. SA increased chlorophyll by 37.61% (pot) and 41.17% (field), and carotenoids by 34.04% (pot) and 38.25% (field), surpassing responses from ABA and ShA. Stress-related parameters showed modest increases, yet SA again proved superior, enhancing $H_2O_2$, proline, POD, and GST by 10.20%, 18.05%, 11.15%, and 20.87% in pot plants, and 9.66%, 7.57%, 10.44%, and 12.92% in the field, respectively. Additionally, SA improved RWC in field-grown plants by 10.16%, underscoring its role in enhancing both photosynthetic and stress resilience pathways even in the absence of drought.

To compare the effects of inducers under drought stress versus drought stress alone, a comparison was made based on the extent to which the inducers influenced various biochemical parameters, relative to the untreated control group, on day 15. Under drought conditions, both in pot (Fig 6B) and field (Fig 6A) experiments, significant reductions in chlorophyll and carotenoids were observed, but induced treatments mitigated these declines. The SA + drought (SAD) treatment restored pigment levels most effectively: 14.65% (pot), 20.28% (field); carotenoids: 12.39% (pot), 17.25% (field). Drought markedly elevated $H_2O_2$ levels (48.48% in pot, 44.35% in field), but SAD minimized this increase to 20.78% and 21.22%, respectively. Antioxidant enzyme activities were also highest in SAD-treated plants, with POD increasing by 141.99% (pot) and 129.39% (field), and GST by 161.26% and 134.80%, respectively. Proline accumulation peaked under SAD (63.46% in pot, 35.25% in field), while RWC decline was least in SAD plants (−17.01% in field), indicating superior water retention.

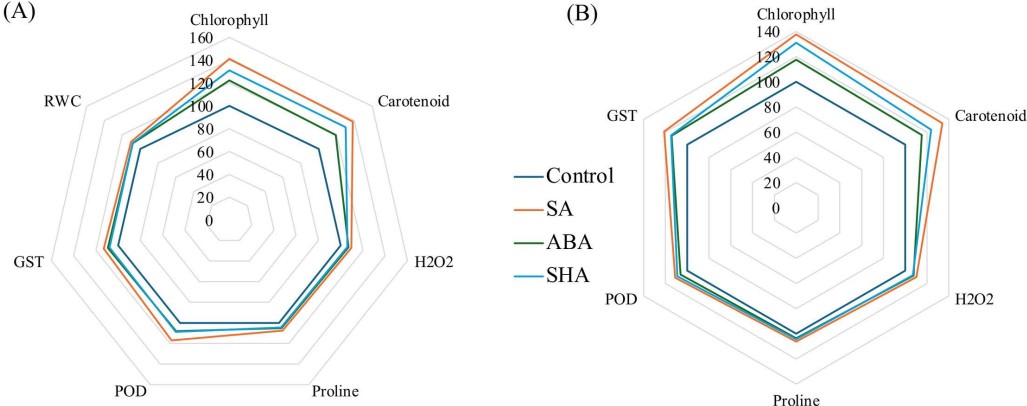

**Fig 5. Comparison of various inducer treatments on physiological and biochemical parameters in plants under normal conditions, relative to well-watered control plants.** The radar chart illustrates the comparative effects of salicylic acid (SA), abscisic acid (ABA), and shikimic acid (ShA) treatments on plants across two conditions: A) Field experiment, and B) Pot Experiment. Each parameter—chlorophyll, carotenoids, $H_2O_2$, proline, peroxidase (POD), and glutathione S-transferase (GST) activity is normalized to the control, set at 100. Values exceeding 100 indicate an increase relative to the control, suggesting enhanced effects under treatment, while values below 100 signify a decrease, indicating reduced effectiveness compared to the control.

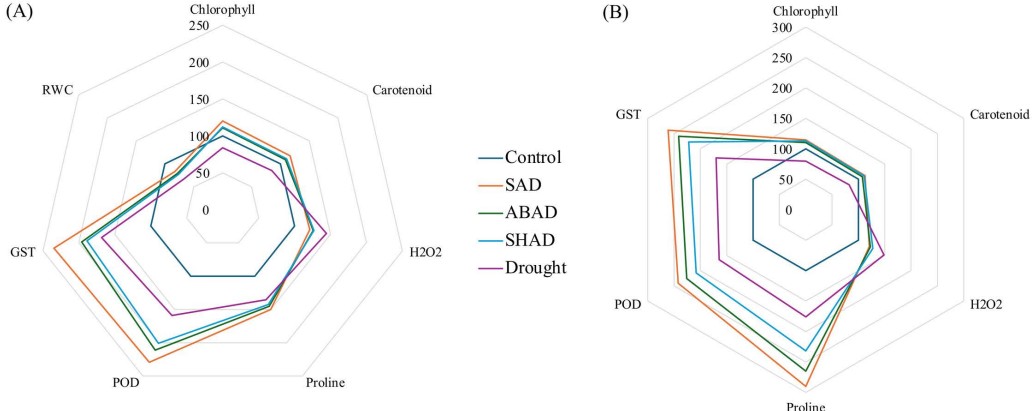

**Fig 6. Comparison of various inducer treatments on physiological and biochemical parameters in plants under drought conditions, relative to both drought-stressed and well-watered control plants.** The radar chart illustrates the comparative effects of salicylic acid (SA), abscisic acid (ABA), and shikimic acid (ShA) treatments on plants across two conditions: A) Field experiment, and B) Pot Experiment. Each parameter, such as chlorophyll, carotenoids, $H_2O_2$, proline, peroxidase (POD), glutathione S-transferase (GST) activity, and RWC, is normalized to the control, set at 100. Values exceeding 100 indicate an increase relative to the control, suggesting enhanced effects under treatment, while values below 100 signify a decrease, indicating reduced effectiveness compared to the control.

## Gene localization and enrichment analysis

To investigate gene functions and associations, Gene Ontology (GO) biological and molecular process analysis, subcellular localization, and KEGG functional enrichment analysis were performed. A total of 76 genes involved in salicylic acid (SA) and shikimic acid (ShA) pathways under drought conditions were identified (S1 Table). The most significant GO terms for biological and molecular processes were analyzed to provide a comprehensive overview of gene functions (Fig 7A). Detailed annotation of the GO enrichment results is provided in (S1 Table).

GO analysis of biological processes revealed that differentially expressed genes (DEGs) were significantly enriched in categories such as activation of response to stimulus, response to chemicals, response to stress, and pigment metabolic process (Fig 7B). For molecular processes, significant enrichment was observed in proline biosynthetic processes, antioxidant activity, inorganic transmembrane transporter activity, regulation of signal transduction, and so on (Fig 8C). Subcellular localization analysis indicated that the DEGs were primarily enriched in plastids, chloroplasts, cytoplasm, and other cellular compartments (Fig 8B). Furthermore, KEGG pathway enrichment analysis showed that DEGs were predominantly involved in pathways mentioning phenylalanine, tyrosine, and tryptophan biosynthesis, glutathione biosynthesis, flavonoid biosynthesis, and carotenoid biosynthesis pathways (Fig 8A).

## Discussion

The current study evaluates the efficacy of salicylic acid, abscisic acid, and shikimic acid in mitigating drought stress in tea plants. Foliar application of these compounds significantly enhanced drought tolerance, as demonstrated by improved biochemical parameters and reduced oxidative damage. Drought stress, recognized as a major limiting factor in agricultural productivity, causes severe physiological disruptions, particularly in photosynthesis, chlorophyll content, and growth parameters [36]. The foliar application of plant growth regulators, such as SA, ABA, and ShA, has been previously studied for their roles in modulating plant responses to abiotic stress [37]. They are involved in critical metabolic pathways, including the glyoxylate cycle and gluconeogenesis, supporting energy metabolism under stress conditions. Their capacity to regulate plant metabolic processes further reinforces their role as a key modulator in stress tolerance [38,39].

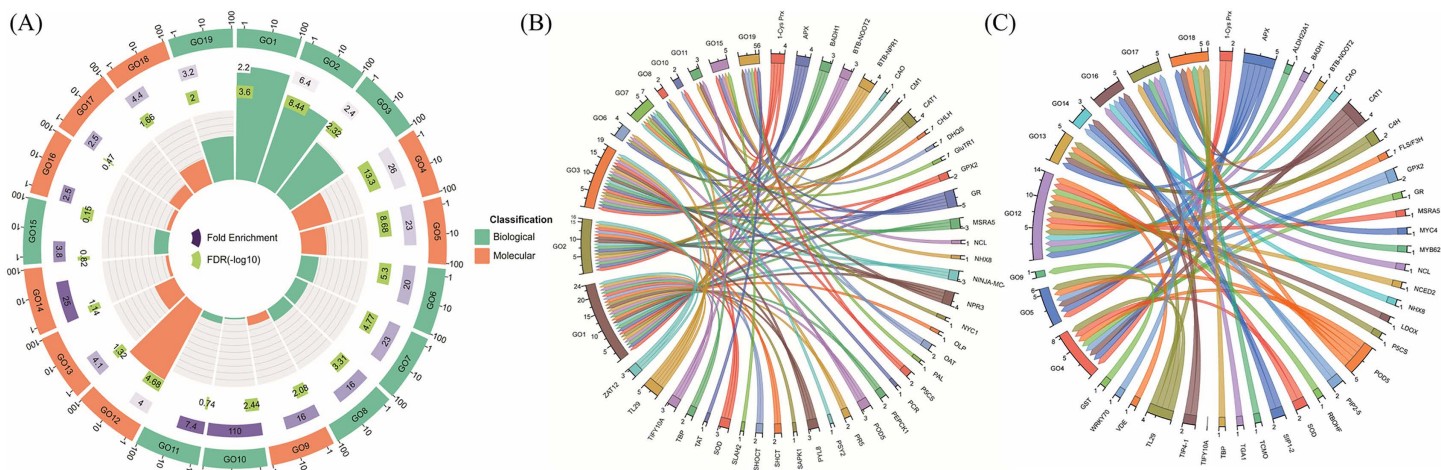

**Fig 7. Distribution of DEGs in GO enrichment analysis.** (A) The order of the circles is from the outside in. The first circle shows the most significantly enriched terms of biological process (BP) and molecular function (MF), with the scale outside corresponding to the number of genes. The second circle shows the number of background genes in the corresponding GO terms and the p-value. The third circle shows the number of DEGs. The inner circle shows the enrichment score of the corresponding GO terms. Genes are involved in SA and ShA signaling pathways with their respective GO terms. (B) Depicts GO terms of Biological Processes (BP), while (C) shows GO terms of Molecular Processes (MP). The direction of the lines indicates gene associations with specific GO terms, and the bar thickness represents the number of genes in each category.

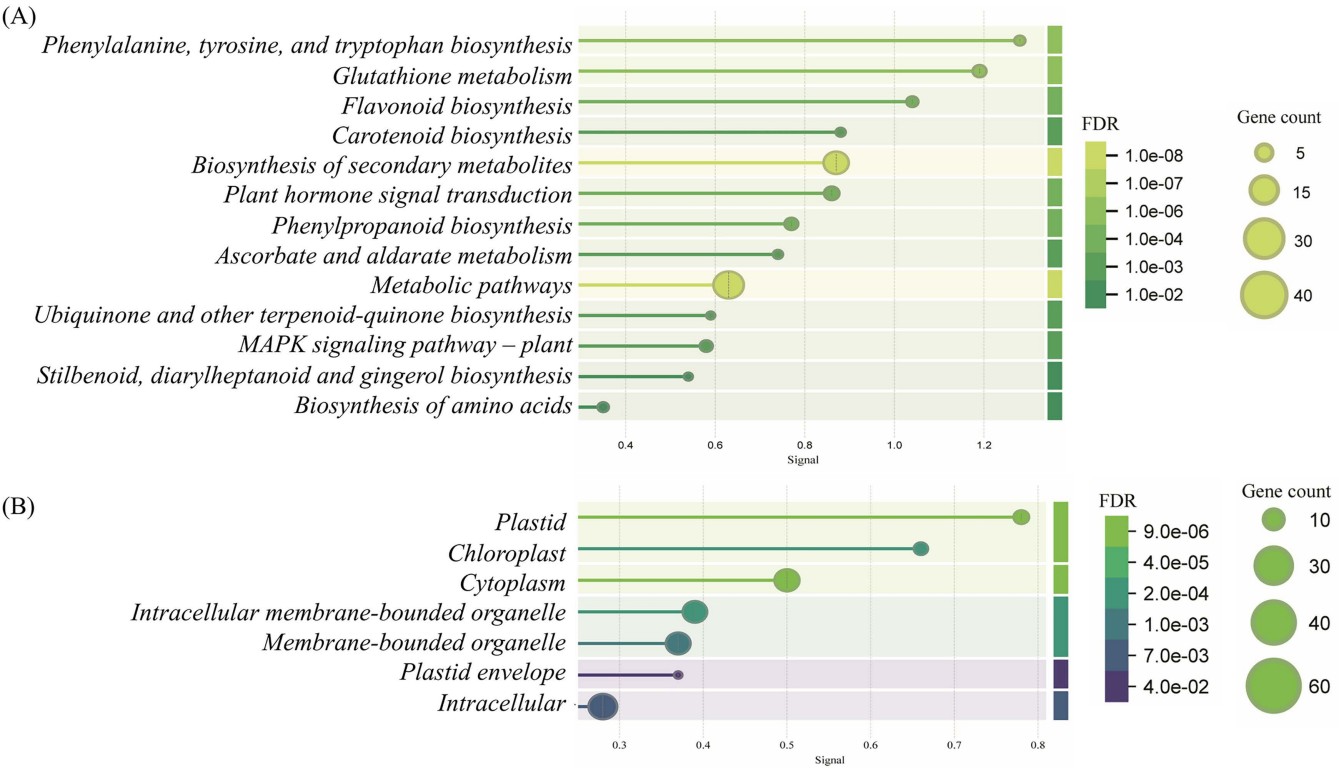

**Fig 8. KEGG pathway and Gene Ontology analysis.** A bubble chart for (A) KEGG pathway and (B) Gene Ontology enrichment analysis term for the category of cellular components of 76 genes identified through a literature review. The Y-axis indicates the pathway and cellular component name, and the X-axis indicates the enriched factor. The bubble size indicates the number of genes, and the color bar indicates the FDR values.

The reduction in chlorophyll content, often observed under drought conditions due to pigment degradation and photosynthetic enzyme impairments, was effectively reduced by the application of SA, ABA, and ShA. The inducers not only mitigate chlorophyll and carotenoid degradation—key indicators of photosynthetic capacity—but also help maintain photosynthesis, as supported by previous literature linking these pigments to photosynthetic efficiency [40,41]. Furthermore, their ability to sustain RWC under field conditions further validates their role in preserving photosynthetic activity under stress (Figs 2A and 2B). This supports earlier findings on the role of exogenous application of SA and ABA in mitigating drought-induced damage by regulating chlorophyll biosynthesis and inhibiting chlorophyll degradation [14,39,42]. A similar outcome was also observed with ShA in a previous study involving tomatoes [27].

Proline, a well-established osmoprotectant, showed significantly elevated accumulation in SA, ABA, and ShA-treated plants under drought conditions (Figs 3A and 3B). This increase in proline content is crucial for maintaining cellular turgor, osmotic balance, and protecting cellular structures during water-deficit stress, aligning with previous studies that proved exogenous SA and ABA applications significantly enhance proline accumulation, thereby improving drought tolerance [17,43]. Similar trends were demonstrated for the ShA treatment in proline metabolism [21,44].

$H_2O_2$ is a common ROS that accumulates during drought-induced oxidative stress. The untreated drought-stressed group showed a significant increase in $H_2O_2$ levels, while plants treated with SA, ABA, or ShA had markedly lower levels, indicating enhanced ROS scavenging (Figs 3C and 3D). This reduction in ROS is consistent with earlier studies [10,45], where the involvement of SA and ABA was reported in modulating antioxidant enzyme activities, particularly superoxide dismutase (SOD), peroxidase (POD), and catalase (CAT). The activities of antioxidant enzymes peroxidase (Figs 4A and 4B) and GST (Figs 4C and 4D) were significantly elevated in treated plants, reflecting an activated antioxidant defence system. The induction of these enzymes is critical in detoxifying ROS and protecting cellular structures from oxidative damage.

The analysis of differentially expressed genes (DEGs) linked to SA and ShA pathways under drought stress emphasizes their role in stress tolerance [46,47]. The genes that are upregulated by SA and ShA treatments under drought stress were enriched in categories such as "response to chemical" and "stress-related response" (Fig 5B). These genes are involved in key processes, including proline accumulation, POD activity, and GST activity. Key GO terms like pigment synthesis, GST activity, and proline biosynthesis share genes with 'response to stimulus', indicating their activation under combined stress and induced conditions (Fig 5C). Subcellular localization indicated that these genes are expressed in plastids, chloroplasts, and cytoplasm, which are the crucial organelles for photosynthetic roles in photosynthesis and metabolic regulations (Fig 6B). KEGG pathway analysis further highlights the involvement of the genes in antioxidant and stress signaling pathways (Fig 6A), aligning with wet lab findings and underscoring the potential of inducers to enhance drought resilience.

The reduction in morphological damage observed in treated plants is due to enhanced activity of antioxidant enzymes and the accumulation of osmoprotectants, which alleviate oxidative stress. Moreover, enhanced biosynthesis of photopigments contributes to improved photosynthetic efficiency and cellular protection. Phytohormones such as SA and ABA have been shown to play critical roles in regulating stomatal closure, thereby minimizing water loss under stress conditions. This stomatal regulation, in turn, helps to conserve water and maintain cellular homeostasis, further protecting the plant from environmental damage [48–50].

Drought stress without inducers reduced photosynthetic pigments despite elevated stress markers, indicating damage to the photosynthetic apparatus. In contrast, the inducers conferred protection, likely through complex molecular mechanisms that enhance drought tolerance. SA enhances stress responses by activating the NPR1-dependent signaling pathway.NPR1 is a key regulator of plant defense mechanisms [51]. In its inactive form, NPR1 exists as an oligomer, stabilized by disulfide bonds.SA accumulation via redox changes triggers a reduction reaction that breaks the disulfide bonds in NPR1, causing NPR1 oligomers to dissociate into active monomers. These monomeric NPR1 proteins can then translocate into the nucleus, where they interact with transcription factors. Together, they bind to the promoters of various

defense-related genes, activating their expression [52]. ShA is a key intermediate in the shikimate pathway, supporting the synthesis of SA and various polyphenolic compounds that function as antioxidants and signaling molecules [53,54]. These polyphenols play a significant role in plant defense mechanisms, acting as antioxidants. Polyphenols can directly engage in defense-related pathways and help in modulating oxidative stress responses [55]. Activation of pathways by internal or external inducers can lead to a moderate increase in H2O2 levels, serving as a signaling molecule to trigger additional defence responses. This balance between H2O2 production and scavenging enables effective stress adaptation while preventing oxidative damage [56,57]. ABA-mediated signaling involves binding to PYR/PYL/RCAR receptors, leading to the inhibition of PP2C phosphatase and activation of SnRK2 kinases. These kinases phosphorylate ABA-responsive transcription factors that induce genes involved in stomatal closure, osmolyte production, and antioxidant responses. PP2C inhibition releases SnRK2 kinases, which subsequently phosphorylate downstream transcription factors (e.g., AREB/ABF). These transcription factors regulate ABA-responsive genes related to stomatal closure, osmolyte synthesis (including proline), and ROS detoxification enzymes such as SOD, POD, and CAT. Additionally, ABA signaling is modulated by chromatin remodeling and the MAPK cascade, which provide transcriptional and posttranscriptional regulation under prolonged stress [45]. Additional layers of regulation involve bZIP transcription factors such as ABP9, histone modifications, and MAPK signaling pathways that further enhance ABA's effectiveness in mitigating drought stress [24]. SA, ShA, and ABA have been shown to enhance drought tolerance in plants; their effects are often transient and may not induce long-term physiological modifications. Evidence from previous studies in spinach showed that SA initially reduced stress responses, but its effectiveness diminished over time. This indicates that those compounds may provide temporary protection without long-term persistence in plant tissues. To sustain their protective role under prolonged drought conditions, repeated application at specific intervals may be required. However, further detailed studies are needed to establish the optimal dosage and potential long-term effects of repeated treatments. Comparative analysis revealed that among the three inducers, SA consistently demonstrated the strongest effect in enhancing drought tolerance. It outperformed ABA and ShA in maintaining chlorophyll content, increasing proline levels, reducing ROS accumulation, and boosting antioxidant enzyme activity. However, ABA contributed mainly through osmotic regulation and stomatal control, while ShA enhanced the synthesis of secondary metabolites and SA precursors, indicating that each compound offers distinct but complementary mechanisms. Future research should incorporate high-throughput transcriptomic and metabolomic approaches to identify key regulatory genes and metabolic pathways involved in drought stress mitigation. Integrating photosynthetic measurements, such as net photosynthetic rate, stomatal conductance, and chlorophyll fluorescence, would also offer a more comprehensive understanding of the effects of SA, ShA, and ABA on plant water relations and photosynthetic performance.

This study provides new insights into the combined and individual roles of SA, ABA, and ShA in enhancing drought tolerance through multiple mechanisms, including enhanced proline synthesis, reduced ROS accumulation, and increased antioxidant enzyme activities. While SA emerged as a particularly potent modulator of drought stress responses, both ABA and ShA also contributed significantly to enhancing plant resilience. These findings not only reaffirm the potential of SA, ABA, and ShA as effective treatments for improving drought tolerance but also provide a mechanistic understanding of how these compounds regulate key metabolic and stress-responsive processes in plants under stress conditions. The antioxidant defense system is crucial for plants' adaptation to drought stress. Several studies have demonstrated that drought-induced oxidative stress in various crops, including *Vigna radiata*, *Brassica napus*, *Sorghum bicolor*, *Zea mays*, and *Solanum lycopersicum*, is closely associated with enhanced antioxidant enzyme activities and increased osmolyte accumulation, both of which contribute to mitigating cellular damage under water-deficient conditions [58–61].

## Conclusion

The study conducted an in-depth comparative analysis of several plant growth regulatory elements, revealing that the application of salicylic acid, abscisic acid, and shikimic acid significantly enhances drought tolerance in tea plants. These regulatory elements bring about beneficial changes in morphological and physiological characteristics, particularly by

enhancing the activity of ROS-scavenging enzymes. These enzymes are vital for detoxifying ROS, which are produced in excess under drought conditions and can lead to cellular damage. By boosting these enzymes, the treatments mitigate oxidative stress and reinforce the plant's overall resilience to drought.

## Supporting information

**S1 Table. Genes involved in the salicylic acid and shikimic acid pathways under drought conditions.**
(XLSX)

## Acknowledgments

The authors extend their gratitude to the Department of Biochemistry and Molecular Biology at Shahjalal University of Science and Technology, Sylhet, Bangladesh, for providing lab facilities and logistic support. The authors also acknowledge the contribution of Lakkatura Tea Estate and Tarapur Tea Estate for providing tea plants and an experimental nursery, respectively.

## Author contributions

**Conceptualization:** Ajit Ghosh.

**Data curation:** Shelina Akter Sheuli, Mir Sultanul Arafin.

**Formal analysis:** Shelina Akter Sheuli, Mir Sultanul Arafin.

**Funding acquisition:** Ajit Ghosh.

**Investigation:** Shelina Akter Sheuli, Mir Sultanul Arafin, Sadia Sultana.

**Methodology:** Shelina Akter Sheuli, Sadia Sultana, Md. Afser Rabbi.

**Project administration:** Ajit Ghosh.

**Resources:** Ajit Ghosh.

**Software:** Md. Afser Rabbi, Ajit Ghosh.

**Supervision:** Ajit Ghosh.

**Validation:** Ajit Ghosh.

**Writing – original draft:** Shelina Akter Sheuli, Mir Sultanul Arafin, Md. Afser Rabbi.

**Writing – review & editing:** Ajit Ghosh.

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
