## [Decision Letter · Decision Letter 0]

6 Mar 2025

PONE-D-25-05165Exogenous Salicylic Acid, Abscisic Acid, and Shikimic Acid Enhance Drought Tolerance in Tea by Modulating Antioxidant Defense and Osmotic RegulationPLOS ONE

Dear Dr. Ghosh,

Thank you for submitting your manuscript to PLOS ONE. After careful consideration, we feel that it has merit but does not fully meet PLOS ONE’s publication criteria as it currently stands. Therefore, we invite you to submit a revised version of the manuscript that addresses the points raised during the review process.

**Dear Authors, After careful reading of the reviewer comments and my own review, there is a lot of scope to improve methods section and discussion. Therefore, I recommend for major revision of the manuscript. **

We look forward to receiving your revised manuscript.

Kind regards,

Ramegowda Venkategowda, PhD

Academic Editor

PLOS ONE

**Journal Requirements:**

Please ensure that your manuscript meets PLOS ONE's style requirements, including those for file naming. The PLOS ONE style templates can be found at https://journals.plos.org/plosone/s/file?id=wjVg/PLOSOne_formatting_sample_main_body.pdf and https://journals.plos.org/plosone/s/file?id=ba62/PLOSOne_formatting_sample_title_authors_affiliations.pdf 2. In your Methods section, please provide additional information regarding the permits you obtained for the work. Please ensure you have included the full name of the authority that approved the field site access and, if no permits were required, a brief statement explaining why. 3. Your ethics statement should only appear in the Methods section of your manuscript. If your ethics statement is written in any section besides the Methods, please move it to the Methods section and delete it from any other section. Please ensure that your ethics statement is included in your manuscript, as the ethics statement entered into the online submission form will not be published alongside your manuscript. 4. Please include captions for your Supporting Information files at the end of your manuscript, and update any in-text citations to match accordingly. Please see our Supporting Information guidelines for more information: http://journals.plos.org/plosone/s/supporting-information.

**Additional Editor Comments:**

Based on the reviewer comments and my own review the methods and discussion has to be improved and hence I recommend major revision.

Reviewers' comments:

Reviewer's Responses to Questions

**Comments to the Author**

1. Is the manuscript technically sound, and do the data support the conclusions?

Reviewer #1: Yes

Reviewer #2: Yes

Reviewer #3: Partly

2. Has the statistical analysis been performed appropriately and rigorously? 

Reviewer #1: Yes

Reviewer #2: Yes

Reviewer #3: Yes

3. Have the authors made all data underlying the findings in their manuscript fully available?

Reviewer #1: Yes

Reviewer #2: Yes

Reviewer #3: Yes

4. Is the manuscript presented in an intelligible fashion and written in standard English?

Reviewer #1: Yes

Reviewer #2: Yes

Reviewer #3: Yes

5. Review Comments to the Author

**Reviewer #1:**  Title; Exogenous Salicylic Acid, Abscisic Acid, and Shikimic Acid Enhance Drought Tolerance in Tea by Modulating Antioxidant Defense and Osmotic Regulation

Journal; PLOS ONE

Manuscript ID; PONE-D-25-05165

Authors; Sheuli et al.,

Date; 02/28/2025

The authors aim to evaluate the effectiveness of three plant growth regulators—salicylic acid (SA), abscisic acid (ABA), and shikimic acid (ShA)—in enhancing drought tolerance in tea plants. Through foliar application of these compounds, the study investigates their impact on biochemical and physiological responses, such as antioxidant enzyme activity, proline accumulation, chlorophyll retention, and oxidative stress mitigation, in order to improve plant resilience under drought conditions. The findings suggest that these compounds can significantly reduce drought-induced damage and enhance drought tolerance, offering potential practical applications for improving agricultural productivity in arid regions.

Overall, the manuscript is well written, the study addresses a crucial agricultural challenge—drought stress—and offers potential solutions for enhancing drought tolerance in tea plants, which could have significant practical applications, especially in water-scarce regions.

Major Concerns;

Can the authors provide details in Materials and Methods section, for the experimental conditions (e.g., concentration of ABA, SA, and ShA, frequency and method of application.

For example, were these compounds applied once, or at specific growth stages? Were there control plants without any treatment?

The manuscript discusses the effects of SA, ABA, and ShA in mitigating drought stress in tea plants, but these findings appear to be based on lab or controlled conditions. It would be important to mention (if not already tested) whether these results would hold up in real-world, field-based conditions. Are the effects as robust outside of the controlled environment? Field validation of these findings would add credibility to the practical implications of the study.

The manuscript describes the positive effects of these treatments on drought tolerance, but it doesn’t address the long-term efficacy of these treatments. How long do the effects last after treatment? Is there any potential for the plants to become reliant on these treatments? The authors can discuss these points.

Although the Discussion highlights key biochemical pathways and compounds, the mechanisms underlying the observed effects (e.g., proline accumulation, ROS scavenging) are mentioned but not explained in sufficient depth. A more in-depth explanation of the molecular mechanisms of SA, ABA, and ShA treatments would strengthen the manuscript, especially for readers unfamiliar with the biochemical pathways involved.

Some parts of the manuscript feel repetitive, especially when discussing the roles of different compounds (SA, ABA, ShA). The same concepts or findings are sometimes restated without adding new information. This could be streamlined to reduce redundancy.

Make sure all abbreviations, such as SA (salicylic acid), ABA (abscisic acid), and ShA (shikimic acid), are introduced fully in the first use. It’s especially important in the Materials and Methods and Discussion sections to make sure the reader is clear on what each acronym represents.

The Conclusion mentions the practical applications of the findings, but it would benefit from suggestions for future research. For example, exploring different concentrations of the compounds, testing them in different crops, or assessing the long-term effects on plant health could offer exciting avenues for further work.

Minor Typos and Phrasing; There are minor language issues that can be addressed, such as;

"capacity of to regulate" should be "capacity to regulate".

"The reduction in morphological damage observed in the induced plants can be attributed to the enhanced activity" could be reworded for clarity. A more concise phrasing could be, "The reduction in morphological damage observed in treated plants is due to enhanced activity".

In some cases, the phrasing in the discussion is slightly repetitive, especially with regard to "ROS accumulation" and "oxidative stress," which could be streamlined.

Thank you,

**Reviewer #2: ** Review comments

I have throughly reviewed the manuscript "Exogenous Salicylic Acid, Abscisic Acid, and Shikimic Acid Enhance Drought

Tolerance in Tea by Modulating Antioxidant Defense and Osmotic Regulation" and highlighted some correction in the manuscript.

1. Abstract

a. Shorten the background.

b. Give values.

c. Give future perspective using one sentence.

2. Introduction

a. Scientific writing should be improved.

b. The introduction could be shortened and rearranged. Mention, the research gap and the aim.

c. Emphasize previous studies

3. Methods

a. Give the coordinates of the collected area.

4. Results

a. Explain results in detail.

b. Data visualization could be improved.

5. Discussion

a. Interrelate results.

b. It is good to relate the results to previously studied. I am sure there may be many available/

c. Give the limitations of the study.

d. What are the recommendations?

e. And future perspectives?

**Reviewer #3: ** The work "Exogenous Salicylic Acid, Abscisic Acid, and Shikimic Acid Enhance Drought Tolerance in Tea by Modulating Antioxidant Defense and Osmotic Regulation" is good and have importance due to increasing need of Tea throughout the world when climate change factor are decreasing the yield of such important crop.

The author have need to carefully remove all formatting problems especially related to references mentioned in text.

But, I will suggest a major changes in this draft because 1) its seems that author have not used the control treatment with water spray to isolate the effect of inducers molecules on tea plant under drought, 2) i will suggest to add the data of photosynthesis and water relation attributes to make the complete pic when you are talking about osmotica and Chl pigments.

Please check the detailed comments on the pdf file.

6. PLOS authors have the option to publish the peer review history of their article (what does this mean? ). If published, this will include your full peer review and any attached files.

**Do you want your identity to be public for this peer review?** For information about this choice, including consent withdrawal, please see our Privacy Policy .

Reviewer #1: No

Reviewer #2: **Yes: ** Zahoor Ahmad Sajid

Reviewer #3: No

---

## [Author Response · Author response to Decision Letter 1]

26 Jun 2025

Reviewer#1

Query 1: Can the authors provide details in the Materials and Methods section, for the experimental conditions (e.g., concentration of ABA, SA, and ShA, frequency, and method of application. For example, were these compounds applied once, or at specific growth stages? Were there control plants without any treatment?

Our Response: We appreciate the reviewer’s constructive comment. In the revised Materials and Methods section, we have now clarified the concentration, timing, and application methodology of the treatments in detail. Specifically, we used 100 µM abscisic acid (ABA), 4 mM salicylic acid (SA), and 4 mM shikimic acid (ShA), which were applied via foliar spraying using a handheld sprayer. Each plant received 15 mL of the respective solution once daily, applied in the early morning for four consecutive days (Days 0–3).

The treatments were administered only once during a defined vegetative growth stage and were not repeated in later developmental stages. Control plants were treated with an equivalent volume (15 mL) of distilled water following the same spraying protocol and were maintained under identical environmental and handling conditions. This information was clearly stated in lines 127–128 of the initial manuscript.

We also acknowledge a previous error in the table describing the control treatment and have corrected it accordingly in the updated version.

Query 2: The manuscript discusses the effects of SA, ABA, and ShA in mitigating drought stress in tea plants, but these findings appear to be based on lab or controlled conditions. It would be important to mention (if not already tested) whether these results would hold up in real-world, field-based conditions. Are the effects as robust outside of the controlled environment? Field validation of these findings would add credibility to the practical implications of the study.

Our Response: We are thankful to the reviewer for highlighting this important point. In response to the suggestion, we conducted a new field-based experiment under natural environmental conditions at the nursery of Tarapur Tea Estate. This experiment was designed to closely replicate the pot trial in terms of treatment groups, concentrations, application timing, and drought duration. However, by introducing natural variability in light, temperature, humidity, and wind, the field setup allowed us to test the reproducibility and robustness of the responses observed in the controlled environment.

To ensure experimental rigor, we used two physically separate raised soil beds to prevent cross-interference between the well-watered and drought-treated groups. A rain-exclusion structure was also implemented to allow drought stress imposition without precipitation interference, following established methodologies.

This field validation now strengthens the translational value of our findings and supports the practical application of SA, ShA, and ABA for drought mitigation in tea cultivation. Details have been added in the revised Materials and Methods section

Query 3: The Conclusion mentions the practical applications of the findings, but it would benefit from suggestions for future research. For example, exploring different concentrations of the compounds, testing them in different crops, or assessing the long-term effects on plant health could offer exciting avenues for further work.

Our Response: We thank the reviewer for this constructive suggestion. In response, we have revised the manuscript to include a paragraph on future perspectives (lines 589–594). Specifically, we now acknowledge that future investigations should incorporate detailed photosynthetic measurements—such as net photosynthetic rate, stomatal conductance, and chlorophyll fluorescence—to provide a more comprehensive understanding of how SA, ShA, and ABA influence plant water relations and photosynthetic efficiency under stress conditions. Furthermore, we emphasize the need for integrated transcriptomic and metabolomic analyses to identify key regulatory genes and metabolic pathways underlying the observed physiological responses. These approaches will enhance mechanistic insights and help optimize the use of these inducers for improving stress tolerance in tea plants.

Query 4: The manuscript describes the positive effects of these treatments on drought tolerance, but it doesn’t address the long-term efficacy of these treatments. How long do the effects last after treatment? Is there any potential for plants to become reliant on these treatments? The authors can discuss these points.

Our Response: We appreciate the reviewer’s insightful comment regarding the long-term efficacy of treatments. In response, we have addressed this point in lines (576 – 583) of the revised manuscript. Specifically, we have discussed the potential for the plants to develop resilience on repeated application of these inducers and the current lack of evidence regarding the duration of their protective effects. To provide context, we included an example from a related study and emphasized the need for future research involving extended field trials and the long-term impact of these treatments.

Query 5: Although the Discussion highlights key biochemical pathways and compounds, the mechanisms underlying the observed effects (e.g., proline accumulation, ROS scavenging) are mentioned but not explained in sufficient depth. A more in-depth explanation of the molecular mechanisms of SA, ABA, and ShA treatments would strengthen the manuscript, especially for readers unfamiliar with the biochemical pathways involved.

Our Response: We thank the reviewer for this valuable suggestion. In response, we have revised the discussion section to provide a detailed explanation of the molecular mechanisms underlying the observed effects, particularly concerning the proline accumulation and ROS scavenging in response to SA, ABA, and ShA treatments. Specifically, we elaborate on the signaling pathways involving NPR1 (for SA), the SnRK2 cascade (for ABA), and the role of ShA as a precursor in the shikimate and phenylpropanoid pathways. The revision can be found on lines (550-575).

Query 6: Minor Typos and Phrasing

Our Response: Thank you for pointing out the minor typographical and phrasing issues. We have carefully reviewed the manuscript and corrected all identified typos, improved sentence clarity, and refined the phrasing to ensure better readability and consistency throughout the manuscript.

Query 7: Some parts of the manuscript feel repetitive, especially when discussing the roles of different compounds (SA, ABA, ShA). The same concepts or findings are sometimes restated without adding new information. This could be streamlined to reduce redundancy.

Our Response: Thank you for the comment. We have revised the manuscript to remove repetitive content and streamline the discussion of SA, ABA, and ShA to improve clarity and avoid redundancy.

Query 7: Make sure all abbreviations, such as SA (salicylic acid), ABA (abscisic acid), and ShA (shikimic acid), are introduced fully in the first use. It’s especially important in the Materials and Methods and Discussion sections to make sure the reader is clear on what each acronym represents.

Our Response: Thank you for the suggestion. We have carefully reviewed the manuscript to ensure that all abbreviations, including SA (salicylic acid), ABA (abscisic acid), and ShA (shikimic acid), are fully defined in all sections of the revised manuscript.

Reviewer#2

Query 1: Introduction:

a. Scientific writing should be improved.

b. The introduction could be shortened and rearranged. Mention, the research gap and the aim.

c. Emphasize previous studies

Our Response: We sincerely thank the reviewer for these insightful comments. In the revised manuscript, we have significantly improved scientific writing in the Introduction section to enhance clarity, readability, and flow. Redundancies have been removed, and sentences have been made more concise and precise, particularly in the discussion of previous studies and relevant literature. We have strengthened the contextualization of our work by emphasizing previous relevant studies.

We have also restructured the Introduction to present a clearer narrative: the background is now more focused, the research gap is explicitly highlighted (lines 96 -100), and the aims and objectives of the study are clearly stated (lines 106–109).

Query 2: Methods

a. Give the coordinates of the collected area

Our Response: We thank the reviewer for this suggestion. In response, we have now clearly provided the geographical coordinates of the experimental location—Lakkatura Tea Estate—in both the pot and field experiment sections of the Materials and Methods. The coordinates have been added as follows: (24.911270°N, 91.888347°E) to ensure precise identification of the experimental site.

Query 3: Result

a. Explain results in detail.

b. Data visualization could be improved.

Our Response: We thank the reviewer for these constructive suggestions. In the revised Results section, we have now provided a more detailed explanation of the data, focusing particularly on the percentage deviations from Day 0 to Day 15 for each parameter. This allows readers to quantitatively assess the magnitude of treatment-induced changes across different experimental groups. Furthermore, we have integrated the description of pot and field experiment outcomes side by side to facilitate direct comparison and help readers appreciate the consistency or variation in response under controlled and field conditions.

To enhance clarity and interpretability, we have improved the data visualization. Specifically, we have separated figures related to photosynthetic pigments and non-enzymatic stress markers to avoid overcrowding and improve visual coherence. Additionally, we have presented radar plots for normal and drought-stressed conditions separately, allowing for a clearer comparison of treatment effects under both conditions.

Query 4: Discussion

a. Interrelate results.

b. It is good to relate the results to previously studied. I am sure there may be many available/

c. Give the limitations of the study.

d. What are the recommendations?

e. And future perspectives?

Our Response: We appreciate the reviewer’s suggestions. The discussion has been revised to better interrelate our findings and connect them with prior studies. We have added a section highlighting the study’s limitations. Recommendations have been included. Additionally, a future perspective has been added. These updates are reflected in the revised manuscript lines from 538 onwards.

Query 5: Abstract

a. Shorten the background.

b. Give values.

c. Give a future perspective using one sentence.

Our Response: Thank you for this suggestion. In response, we have made the necessary changes, and these changes can be found in the revised abstract section.

Reviewer#3

Query 1: It seems that the authors have not used a control treatment with water spray to isolate the effect of induced molecules on tea plants under drought.

Our Response: We thank the reviewer for this important observation. To clarify, control plants were indeed included and were treated with an equivalent volume (15 mL) of distilled water via foliar spraying, following the same application schedule (once daily from Day 0 to Day 3) as the treatment groups. These control plants were maintained under identical environmental and handling conditions across both pot and field experiments to ensure experimental consistency and isolate the effects of the induced molecules. This information was explicitly stated in lines 127–128 of the initial manuscript.

We also acknowledge that the original treatment table contained an error regarding control treatment labeling. This has now been corrected in the updated version for better clarity and transparency.

Query 2: I would suggest adding data on photosynthetic and water relation attributes to provide a complete picture when discussing osmotica and chlorophyll pigments.

Our Response: We appreciate the reviewer’s insightful suggestion. In the initial version of the manuscript, we aimed to infer the status of photosynthesis and water relations based on changes in chlorophyll and carotenoid contents. As these pigments are directly involved in light harvesting and photoprotection, their stability under drought stress often reflects preserved photosynthetic potential. However, we agree that such inferences remain indirect without supporting physiological measures.

In response to this comment, we have now incorporated Relative Water Content (RWC) data from the field experiment, as a physiological indicator of water status under drought stress (see revised result section, lines 356-370 and Figure 3E). Unfortunately, RWC measurements were not conducted during the pot experiment, as this parameter was not included in the original design.

Additionally, we acknowledge that direct measurements of photosynthetic parameters (e.g., gas exchange, Fv/Fm) would have strengthened the interpretation. However, due to equipment limitations, such data could not be obtained in the present study. We have now clearly stated this limitation in the revised discussion section (lines 591-594) and suggested that future studies should include photosynthetic performance assessments to complement pigment analysis and provide a more comprehensive understanding.

We believe the inclusion of RWC data enhances the physiological interpretation and better supports the conclusions drawn about drought resilience.

Query 3: The Format needs to be checked carefully.

Our Response: Thank you for your comment. We have thoroughly revised the manuscript to ensure consistent formatting throughout the Introduction, Methods, and Discussion sections.

Query 4: Please check the detailed comments on the PDF file

Our Response: Thank you for your detailed comments provided in the PDF file. We have thoroughly reviewed each annotation and incorporated the suggested corrections and clarifications into the revised manuscript. All changes have been addressed to improve the overall clarity, accuracy, and quality of the manuscript.

---

## [Decision Letter · Decision Letter 1]

17 Aug 2025

Exogenous Salicylic Acid, Abscisic Acid, and Shikimic Acid Enhance Drought Tolerance in Tea by Modulating Antioxidant Defense and Osmotic Regulation

PONE-D-25-05165R1

Dear Dr. Ghosh,

We’re pleased to inform you that your manuscript has been judged scientifically suitable for publication and will be formally accepted for publication once it meets all outstanding technical requirements.

Kind regards,

Ramegowda Venkategowda, PhD

Academic Editor

PLOS ONE

Additional Editor Comments (optional):

Reviewers' comments:

Reviewer's Responses to Questions

**Comments to the Author**

1. If the authors have adequately addressed your comments raised in a previous round of review and you feel that this manuscript is now acceptable for publication, you may indicate that here to bypass the “Comments to the Author” section, enter your conflict of interest statement in the “Confidential to Editor” section, and submit your "Accept" recommendation.

Reviewer #1: All comments have been addressed

Reviewer #2: All comments have been addressed

2. Is the manuscript technically sound, and do the data support the conclusions?

Reviewer #1: Yes

Reviewer #2: Yes

3. Has the statistical analysis been performed appropriately and rigorously? 

Reviewer #1: I Don't Know

Reviewer #2: Yes

4. Have the authors made all data underlying the findings in their manuscript fully available?

Reviewer #1: Yes

Reviewer #2: Yes

5. Is the manuscript presented in an intelligible fashion and written in standard English?

Reviewer #1: Yes

Reviewer #2: Yes

6. Review Comments to the Author

Reviewer #1: The replies to the reviewers comments and the corrections made are justified. I do not have any other questions on the manuscript. Thank you.

Reviewer #2: It seems that authors have done all the changes suggested by the reviewers. So manuscript may be accepted.

7. PLOS authors have the option to publish the peer review history of their article (what does this mean? ). If published, this will include your full peer review and any attached files.

**Do you want your identity to be public for this peer review?** For information about this choice, including consent withdrawal, please see our Privacy Policy .

Reviewer #1: No

Reviewer #2: **Yes: ** Dr. Zahoor Ahmad Sajid

---

## [Editor Report · Acceptance letter]

PONE-D-25-05165R1

PLOS ONE

Dear Dr. Ghosh,

I'm pleased to inform you that your manuscript has been deemed suitable for publication in PLOS ONE. Congratulations! Your manuscript is now being handed over to our production team.

Kind regards,

on behalf of

Dr. Ramegowda Venkategowda

Academic Editor

PLOS ONE